# Melanopsin Carboxy-terminus phosphorylation plasticity and bulk negative charge, not strict site specificity, achieves phototransduction deactivation

**Juan C. Valdez-Lopez**[1], **Sahil Gulati**[2,3], **Elelbin A. Ortiz**[1,4], **Krzysztof Palczewski**[2], **Phyllis R. Robinson**[1]*

**1** Department of Biological Sciences, University of Maryland Baltimore County, Baltimore, Maryland, United States of America, **2** Department of Ophthalmology, Gavin Herbert Eye Institute, University of California, Irvine, California, United States of America, **3** Gatan Inc, Pleasanton, California, United States of America, **4** Department of Cell and Developmental Biology, Perelman School of Medicine, University of Pennsylvania, Philadelphia, Pennsylvania, United States of America

* probinso@umbc.edu

**Data Availability Statement:** All mass spectrometry files can be found on as datasets on MassIVE using the following URLs: ftp://massive.

## Abstract

Melanopsin is a visual pigment expressed in a small subset of ganglion cells in the mammalian retina known as intrinsically photosensitive retinal ganglion cells (ipRGCs) and is implicated in regulating non-image forming functions such as circadian photoentrainment and pupil constriction and contrast sensitivity in image formation. Mouse melanopsin's Carboxy-terminus (C-terminus) possesses 38 serine and threonine residues, which can potentially serve as phosphorylation sites for a G-protein Receptor Kinase (GRK) and be involved in the deactivation of signal transduction. Previous studies suggest that S388, T389, S391, S392, S394, S395 on the proximal region of the C-terminus of mouse melanopsin are necessary for melanopsin deactivation. We expressed a series of mouse melanopsin C-terminal mutants in HEK293 cells and using calcium imaging, and we found that the necessary cluster of six serine and threonine residues, while being critical, are insufficient for proper melanopsin deactivation. Interestingly, the additional six serine and threonine residues adjacent to the required six sites, in either proximal or distal direction, are capable of restoring wild-type deactivation of melanopsin. These findings suggest an element of plasticity in the molecular basis of melanopsin phosphorylation and deactivation. In addition, C-terminal chimeric mutants and molecular modeling studies support the idea that the initial steps of deactivation and β-arrestin binding are centered around these critical phosphorylation sites (S388-S395). The degree of functional versatility described in this study, along with ipRGC biophysical heterogeneity and the possible use of multiple signal transduction cascades, might contribute to the diverse ipRGC light responses for use in non-image and image forming behaviors, even though all six sub types of ipRGCs express the same melanopsin gene OPN4.

ucsd.edu/MSV000084457/ and ftp://massive.ucsd.edu/MSV000084461/.

**Funding:** NIH Training Grant T32 GM066706 awarded to J.C.V.L.,NIH R01 EY027202-01A1 awarded to P.R.R., NIH shared instrument grant 1S10OD023436-01 (purchase of the Fusion Lumos tribrid mass spectrometer system), and the and the Alcon Research Institute (ARI; to K.P.) and Research to Prevent Blindness to The Department of Ophthalmology at UCI. K.P. is the Leopold Chair of Ophthalmology at the University of California, Irvine. The funders had no role in study design, data collection and analysis, decision to publish, or preparation of the manuscript. Gatan Inc. is the current location for author SG, and data collection was done while affiliated with the University of California, Irvine. Gatan Inc. currently supports in the form of salaries for author SG, but did not have any additional role in the study design, data collection and analysis, decision to publish, or preparation of the manuscript. The specific roles of this author are articulated in the "Author Contributions" section.

**Competing interests:** The authors declare that they have no conflicts of interest with the contents of this article. Affiliation with Gatan Inc. presents no conflict of interest with the contents of this article; it is the current affiliation of author SG. This does not alter our adherence to PLOS ONE policies on sharing data and materials.

## Introduction

Visual pigments are light-sensitive molecules comprised of specialized G protein-coupled receptors (GPCRs), or opsins, which are covalently linked to a retinaldehyde molecule, typically 11-*cis*-retinal in the mammalian retina. Upon absorption of a photon of light, retinal photoisomerizes and triggers a conformational change in the visual pigment, and subsequently activates a G-protein signaling cascade, thus converting light into an intracellular signal. Melanopsin is a unique visual pigment expressed in intrinsically photosensitive retinal ganglion cells (ipRGCs) in mammals [1, 2, 3]. IpRGCs represent a small population of the total cells in the ganglion cell layer. These cells are the primary regulators of non-image forming processes such as light-induced pupil constriction, circadian photoentrainment, and sleep [4, 2]. Based on sequence homology, melanopsin is more closely related to visual pigments found in rhabdomeric photoreceptors found in organisms such as *Drosophila* [5, 1]. Specifically, mouse melanopsin shares 27% amino acid sequence identity with mouse rhodopsin, a ciliary (C-type) opsin while sharing 31.5% identity with squid rhodopsin, a rhabdomeric (R-type) opsin. Thus, melanopsin is predicted to function and signal in a manner distinct from mammalian visual pigments expressed in rod and cone photoreceptors. Compared to classical image-forming rod and cone photoreceptor cells in the mammalian retina, ipRGCs exhibit sluggish light responses [3], characterized by a unique capability to sustain a light response for extended periods of time [6, 7]. The basis of these unique ipRGC light-response kinetics might be attributed to molecular features of the visual pigment that prolong its desensitization, such as its proposed bi- or possible tri-stability [7], which would prevent photobleaching of the visual pigment and thus support sustained light responses. Additionally, melanopsin's proposed lower affinity for the chromophore compared to rhodopsin, particularly in diurnal mammals [8], could contribute to the ipRGC's lower light sensitivity and sluggish kinetics, compared to rod and cone photoreceptor cells.

In M1-type ipRGCs, light-activated melanopsin is proposed to couple to Gαq/11 [9, 10, 11], which activates PLCβ4 that ultimately results in a depolarization of the cell. Previous studies suggest that melanopsin signaling desensitization is initiated by G protein-coupled Receptor Kinase (GRK)-mediated C-terminal phosphorylation [12, 13, 14, 15] which stimulates β-arrestin binding [16, 17]. However, there is evidence suggesting that GRK2-mediated phosphorylation is not critical in ipRGCs [18], and other studies also suggest that melanopsin phosphorylation can be mediated by alternate kinases such as Protein Kinase C [19], S6K1 [19], and Protein Kinase A [20]. These data suggest diverse molecular regulation of melanopsin phosphorylation, unlike the canonical GRK-mediated phosphorylation of GPCRs. Additional analysis of mouse melanopsin deactivation via calcium assays in tissue culture cells suggests that a cluster of six C-terminal residues, S388, T389, S391, S392, S394, S395, are required to deactivate melanopsin signaling [13], and similar mutagenesis of C-terminal phosphorylation sites, S381, S384, T385, S388, T389, S391, S392, S394, S395, significantly affects melanopsin desensitization *in vivo* as well [14]. C-terminus truncation experiments also support the importance of these putative phosphorylation sites in signaling deactivation [13, 14]. Previous studies by Fahrenkrug et al (2014) [19] in rat melanopsin suggest that residues S381 and S398 are likely phosphorylated and they also are important in regulation of light-induced calcium responses. Furthermore, transgenic mice expressing melanopsin with mutated C-terminal phosphorylation sites greatly prolong the kinetics of ipRGC and pupillary light responses [14, 15].

Melanopsin's C-terminus is unique amongst mammalian opsins. It is 171 amino acids long and comprises 38 serine and threonine residues (Fig 1), which are potential sites for phosphorylation. This contrasts with the C-terminus of the image-forming visual pigment rhodopsin,

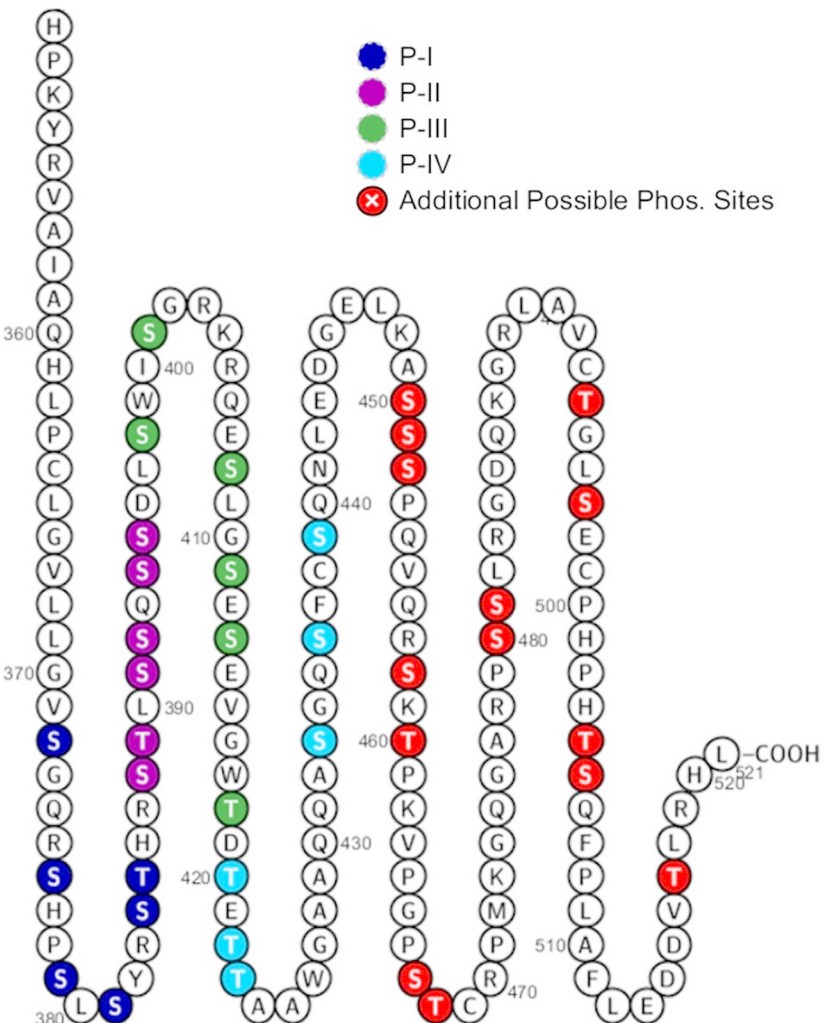

**Fig 1. Melanopsin has 38 putative sites of C-terminus phosphorylation.** Serine and threonine residues are represented by colored circles. Clusters of six putative phosphorylation sites of interest, denoted as P-I, P-II, P-III, P-IV, are in different colors. The proximal C-terminus is defined as the region that includes P-I and P-II, while the distal C-terminus includes P-III, P-IV, and beyond. Figure made using Protter [52].

which is 39 amino acids long and has 8 possible sites for phosphorylation. Mass spectrometric analysis of rhodopsin suggests site-specific phosphorylation of its C-terminus at three serine and threonine residues in response to light [21, 22, 23, 24, 25]. A similar number of phosphorylations was reported in the neuropeptide FF2 receptor [26], and up to six phosphorylation sites were identified in the β2-adrenergic receptor [27, 28, 29]. Here, we hypothesize that the proper function of mouse melanopsin's C-terminus during deactivation requires a larger bulk negative charge due to more serine and threonine phosphorylations than in opsins expressed in rod and cone photoreceptors. This requirement for extra phosphorylation sites in the C-terminus may contribute to melanopsin's sustained and slow light response. We found that a bulk negative charge on melanopsin's C-terminus is centered at P-II (The second cluster of putative C-terminal phosphorylation sites, see Fig 1) on the proximal region, but neighboring serine and threonine residues are necessary and capable of contributing to melanopsin deactivation as well. This suggests a unique level of flexibility in the capability of melanopsin's

C-terminus to regulate signaling deactivation through phosphorylation, and this property paves the way to understanding the unique signaling kinetics of the ipRGC.

## Results

### Six putative phosphorylation sites, P-II, are necessary but insufficient to deactivate melanopsin

One approach employed by previous studies to identify critical sites of phosphorylation was the mutation of predicted GRK phosphorylation sites on wild-type melanopsin. Here, we initially took a similar approach by examining the importance of proximal C-terminus serine and threonine residues through mutagenesis of clusters of six sites at a time (Fig 1). Our data suggest that the first six serine and threonine residues on the C-terminus, P-I (S372, S376, S379, S381, S384, T385), are not required to deactivate melanopsin (Fig 2), when tested using a calcium imaging assay of transiently transfected HEK293 cells. Interestingly, mutation of P-I (P-I Null) does cause an apparent reduction in the activation kinetics (Fig 2), thus suggesting a potential role of the C-terminus in melanopsin activation rather than deactivation. Mutagenesis of the next downstream cluster of six sites, P-II (S388, T389, S391, S392, S394, S395) (P-II Null), reduced the rate of melanopsin deactivation, similar in magnitude to the phosphonull melanopsin mutant, which features mutations at all 38 possible sites of C-terminus phosphorylation to alanine residues. In addition, this result is also consistent with the findings described in Blasic et al (2014) [13], where mutagenesis of those same six serine and threonine residues or their elimination via C-terminus truncation resulted in prolonged melanopsin deactivation. We also synthesized melanopsin mutants where the distally located putative sites of phosphorylation, P-III (S398, S401, S408, S411, S413, T418) and P-IV (T420, T422, T423, S433, S436, S439) (P-III Null and P-IV Null), were mutated to alanines. Mutation of the P-III cluster produces a sizable and significant reduction in the deactivation rate of melanopsin, but the extent of the reduction is less than the P-II mutant. Interestingly, mutation of the P-IV cluster also results in a very slight, but significant defect in deactivation rate compared to wild-type melanopsin. These experiments support the importance of the P-II cluster as the most functionally

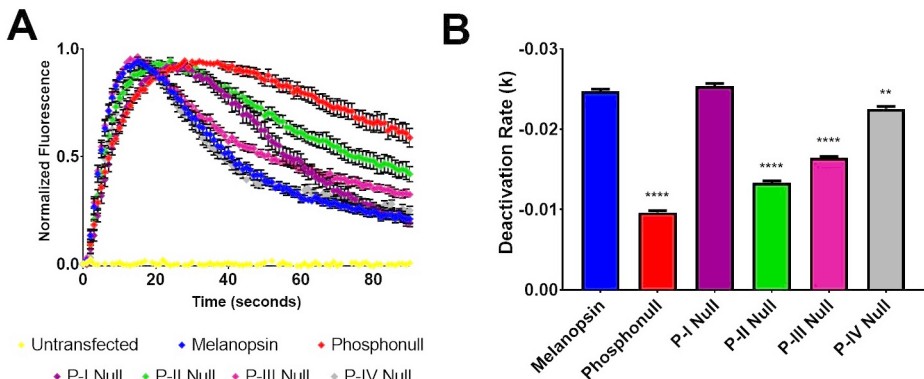

**Fig 2. Proximal phosphorylation sites show differing degrees of importance in melanopsin deactivation.** (A) Calcium imaging assay of HEK293 cells transiently expressing melanopsin and melanopsin mutants with either P-I, P-II, P-III, P-IV, or all 38 serine and threonine residues (phosphonull) mutated to alanines. Mutation of P-II results in the most severe defective deactivation kinetics, and P-I is not critical to the deactivation response. Error bars represent S.E.M. of all transfections pooled together. (B) Quantification of deactivation rates, error bars represent S.E.M. of three independent transfections. All rates compared to wild-type melanopsin's rate of deactivation. P<0.05, 0.01, 0.001, 0.0001 correspond to *, **, ***, ****, respectively.

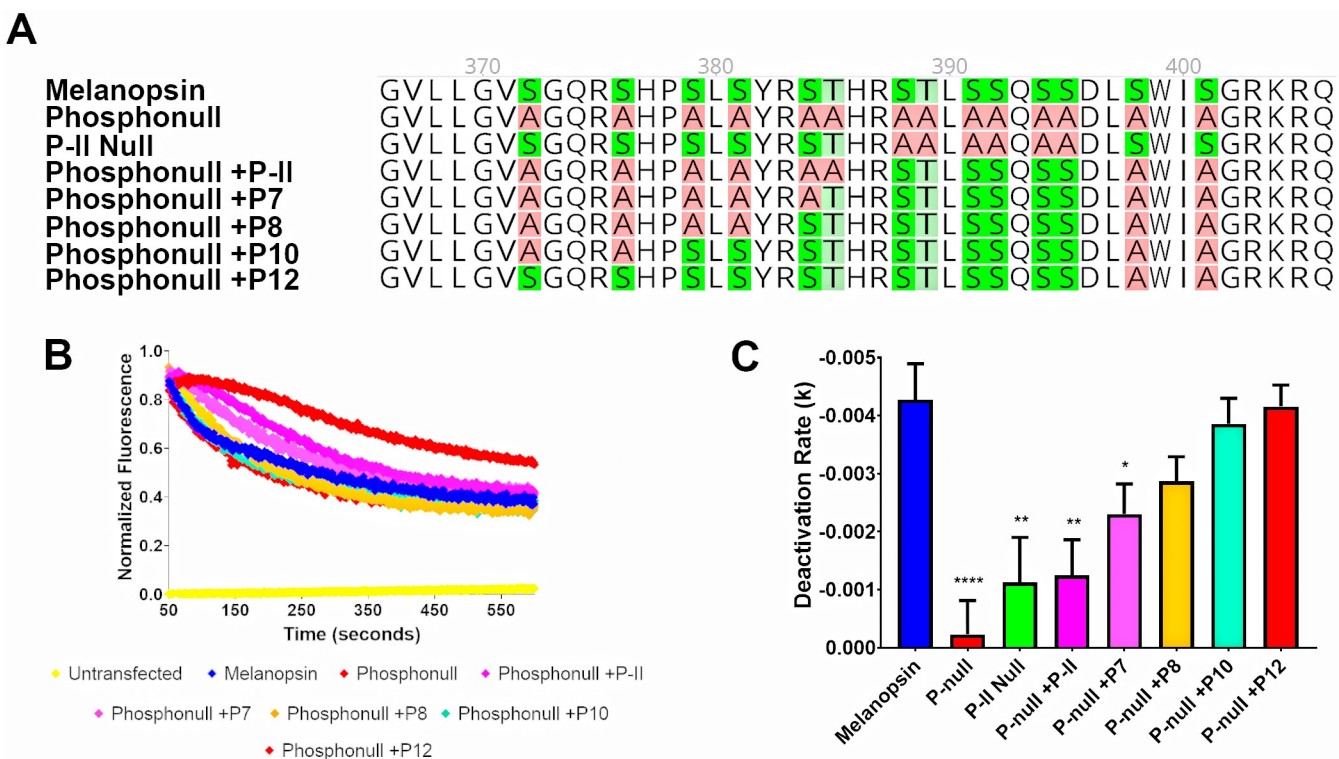

**Fig 3. At least eight proximal C-terminus phosphorylation sites are needed to deactivate melanopsin.** (A) 2D schematic of melanopsin and phosphonull mutants analyzed in this figure. (B) Representative calcium imaging of transient melanopsin-expressing HEK293 cells expressing melanopsin and phosphonull mutants with varying numbers of proximal C-terminus phosphorylation sites mutated back to serines or threonines. (C) Quantified deactivation rates of all melanopsin mutants examined in this experiment. Error bars represent S.E.M. of three independent transfections. All mutants' rates were compared to wild-type melanopsin's rate. P<0.05, 0.01, 0.001, 0.0001 correspond to *, **, ***, ****, respectively. 2D schematic of melanopsin sequences made using Geneious software [53].

significant serine and threonine cluster on the C-terminus, but also suggests a functional role of the distal C-terminus potential phosphorylation sites.

We next employed an alternative approach to examine C-terminus phosphorylation: using the phosphonull melanopsin mutant, select putative phosphorylation sites of interest were mutated back to the serine or threonine residues as in wild-type mouse melanopsin. In doing so, we better describe which phosphorylation sites are sufficient for the deactivation of melanopsin and also test which serines and threonines are likely to be phosphorylated in wild-type melanopsin. First, we tested if P-II is sufficient to deactivate melanopsin (Fig 3) by mutating those sites to serines and threonines in the phosphonull melanopsin construct (phosphonull + P-II). Calcium imaging of this mutant suggests that the presence of this cluster of six putative phosphorylation sites is insufficient to observe deactivation of melanopsin (Fig 3), despite it being necessary (Fig 2) due to the phosphonull + P-II mutant exhibiting reduced deactivation kinetics similar to the P-II Null mutant.

## P-II and additional putative phosphorylation sites in the proximal region of the C-terminus, P-I, are needed to properly deactivate melanopsin

To determine how many phosphorylation sites are needed to deactivate melanopsin, more putative phosphorylation sites were mutated from alanine to serine/threonine using the phosphonull +P-II mutant as the starting template. While the most proximal cluster of six

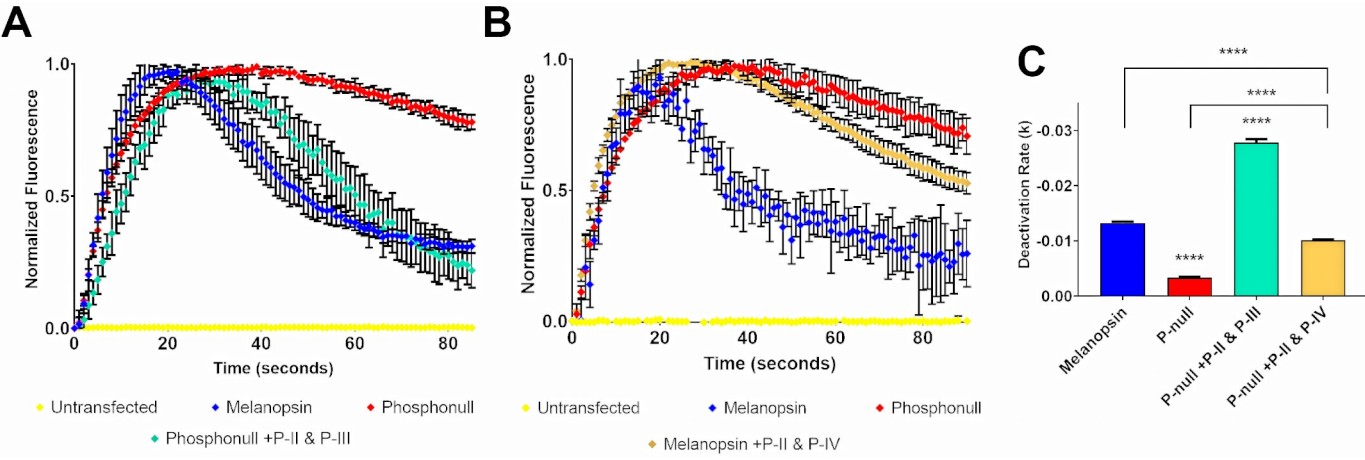

**Fig 4. Distal C-terminus phosphorylation sites can deactivate melanopsin, to varying degrees.** (A) Representative calcium imaging of phosphonull mutants with twelve phosphorylation sites mutated from alanine residues to serine or threonine residues, specifically, P-II and P-III. Error bars represent the S.D. (B) Representative calcium imaging of a phosphonull mutant, analyzing P-II and P-IV. Error bars represent the S.D. (C) Quantification of deactivation rates of melanopsin mutants analyzed in (A) and (B). Error bars in (C) represent the S.E.M. of three independent transfections. P<0.05, 0.01, 0.001, 0.0001 correspond to *, **, ***, ****, respectively.

phosphorylation sites, P-I, are not required for melanopsin deactivation, we tested if they played any functional role in this process (Fig 3). We tested a series of melanopsin phosphonull mutants in a calcium imaging assay using transiently transfected HEK293 cells. Our findings suggest that at least eight of the proximal twelve possible phosphorylation sites are needed to deactivate melanopsin (Fig 3). Specifically, the required cluster, P-II, is needed in conjunction with the proximal cluster P-I. Additional mutants tested (S1 Fig) suggest that P-I alone is also insufficient to deactivate melanopsin. Furthermore, putative phosphorylation sites in the proximal cluster closer to P-II, particularly S384 and T385, deactivate melanopsin at a faster rate compared to the proximal residues in this cluster (S2 Fig).

## P-II and additional putative phosphorylation sites located distally on the C-terminus can also deactivate melanopsin

Truncation of mouse melanopsin's C-terminus after P-II or beyond produces no defect in melanopsin's signaling kinetics when tested in a calcium imaging assay [13]. This experiment suggests that there is no role of the distal C-terminus in regulating melanopsin deactivation, and therefore suggesting that phosphorylation of the distal C-terminal amino acids is not necessary. Using a similar approach as the previous experiments, we synthesized phosphonull mutants where P-II residues were mutated from alanine to serine/threonine. Additionally, two clusters of six putative phosphorylation sites downstream on the C-terminus (P-III or P-IV, Fig 4) were also mutated in a similar manner. Expression and calcium imaging of these mutants suggests that the distal C-terminus phosphorylation sites along with the P-II region can indeed deactivate melanopsin (Fig 4). Specifically, P-II and P-III deactivate melanopsin (Phosphonull +P12 (P-II, P-III), Fig 4). Interestingly, analysis of the deactivation rate suggests that this mutant deactivates melanopsin at a faster rate than wild-type melanopsin (Fig 4), suggesting once again that the possible phosphorylation sites immediately adjacent to the required cluster of six sites are important for melanopsin deactivation. The difference in deactivation rates between the wildtype melanopsin and the phosphonull +P12 (P-II, P-III) mutant might not be readily apparent in Fig 4A due to the extended nature of the mutant's deactivation

kinetics that results in a lower plateau at the end of the assay. However, when the data was fitted to an exponential decay function the decay rate of the mutant is faster than wildtype melanopsin's deactivation rate.

Next, we tested if P-IV can also contribute to melanopsin deactivation by using the phosphonull construct to mutate P-II and P-IV (Phosphonull +12 (P-II, P-IV), Fig 4) to serines and threonines. This resulted in a moderate, yet significant improvement of the deactivation rate compared to phosphonull (Fig 4). However, these mutations were insufficient to fully deactivate melanopsin (Fig 4), suggesting that the distal region of the C-terminus can indeed play a role in this process, but confirming that the required six putative phosphorylation sites and the putative phosphorylation sites surrounding this cluster are the most important in deactivating melanopsin.

## Melanopsin chimeric mutants further support the importance of proximal C-terminus phosphorylation sites in deactivating melanopsin

To further test the importance of the phosphorylation sites' location in deactivating melanopsin, we synthesized melanopsin C-terminus chimeras by fusing truncated melanopsin with the C-termini of β-adrenergic receptor or angiotensin II type-1a receptor. Melanopsin was truncated directly before P-II, at residue R387, and the chimeric C-termini contain the important phosphorylation sites of their respective receptors. In doing so, we further tested the importance of phosphorylation sites at the P-II region, which is compensated for in the β-adrenergic receptor chimera (Fig 5), but not in the angiotensin II type-1a receptor (Fig 5). This is due to the melanopsin-β-adrenergic receptor chimera containing phosphorylation sites located immediately following the melanopsin truncation, and not so in the melanopsin-angiotensin II type-1a receptor chimera. Upon testing these chimeras using calcium imaging, the melanopsin-β-adrenergic receptor chimera showed signaling kinetics that closely mimics wild-type melanopsin (Fig 5). On the other hand, the melanopsin-angiotensin II type-1a receptor's deactivation kinetics are much slower and keeps the chimeric receptor active for a prolonged duration (Fig 5). This further supports the idea that melanopsin requires serine and threonine residues at both the location of P-II, and adjacent to it.

## Phospho-mimetic mutations in phosphonull melanopsin do not rescue the delayed deactivation rate

To further test the importance of proximal C-terminus phosphorylation sites, we designed a phospho-mimetic mutant using phosphonull melanopsin as a template. P-I and P-II were mutated from alanines to aspartic acids (phosphonull +P12 (P-ID, P-IID)) and tested using calcium imaging (Fig 6). Likely, deactivation of the phosphomimetic-mutated-phosphonull through phosphorylation-independent and charge-based coupling to β-arrestin is predicted to occur as previously shown in the chemokine receptor D6 [30]. However, not only did the phosphomimetic mutant fail to rescue phosphonull melanopsin's deactivation defect, the mutant displayed a modulated activation (Fig 6). Specifically, it failed to reach peak signaling intensity during the time course of the experiment, thus suggesting a reduced capability of this mutant to activate the signaling transduction cascade, possibly through the formation of multiple salt bridges involving the proximal region of the C-terminus due to the introduction of many charged residues. It is also possible that this phosphomimetic mutant could rapidly, but reversibly bind β-arrestin, potentially prior to light activation, thus reducing the activation rate. Similar to the data in Fig 2a, the phosphomimetic data suggest other regulatory roles for the C-terminus aside from signaling deactivation.

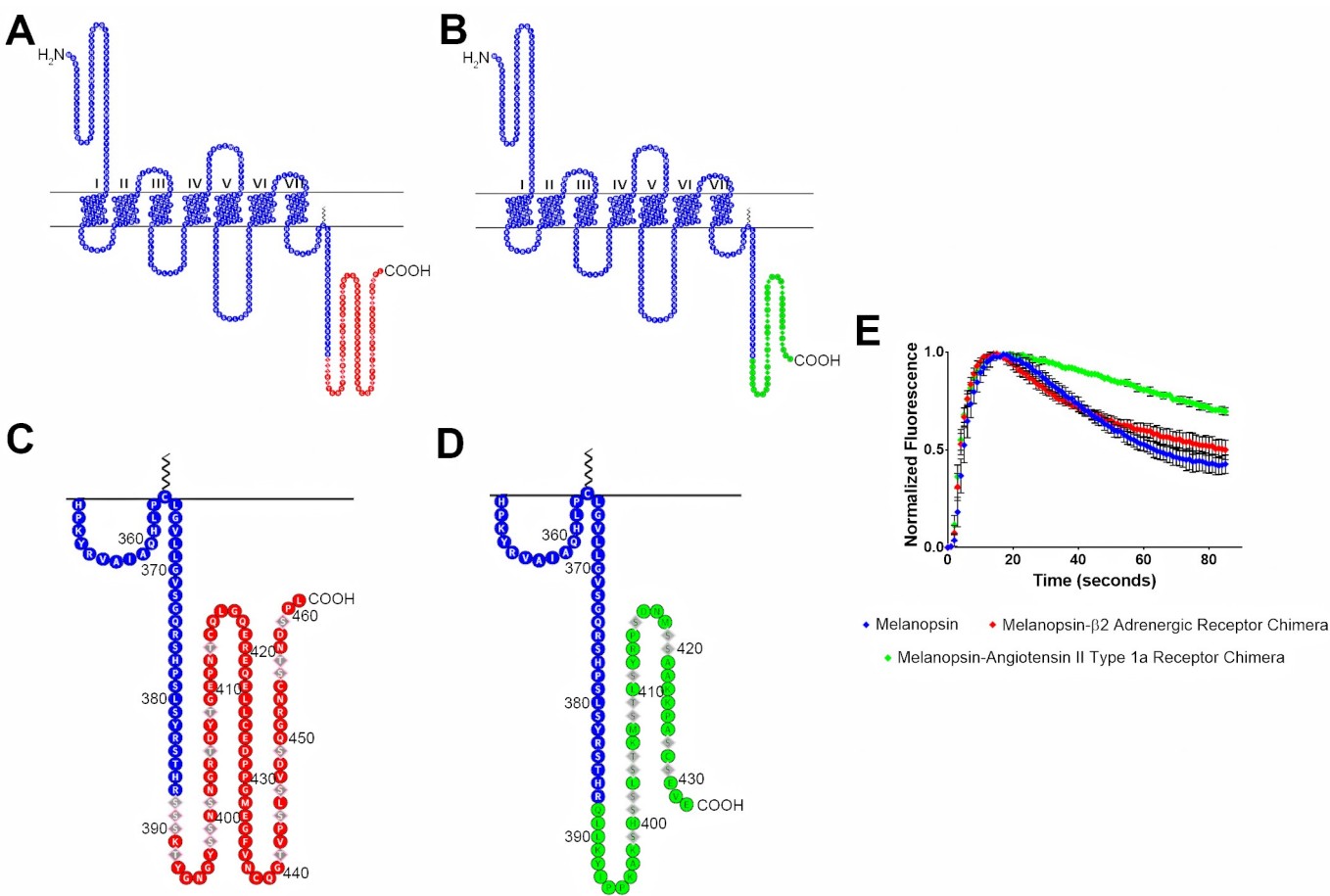

**Fig 5. Melanopsin—β2-adrenergic receptor and melanopsin—Angiotensin II type-1a receptor chimeric mutants display different deactivation kinetics.** In both chimeric constructs, melanopsin was truncated at residue 387, which is right before P-II. (A) 2D schematic of melanopsin-β$_2$-adrenergic receptor chimera, melanopsin residues are represented in blue and β$_2$AR residues are represented in red. Phosphorylation sites on β$_2$AR are represented by diamond shaped residues. (B) 2D schematic of melanopsin-angiotensin II type-1a receptor chimera, melanopsin residues are represented in blue and ATII1aR residues are represented in green. Phosphorylation sites are represented by diamond shaped residues. (C & D) Zoomed in schematics of chimeric C-termini of melanopsin-β$_2$-adrenergic receptor chimera and melanopsin-angiotensin II type-1a receptor chimera. (E) Calcium imaging of chimeric mutants, error bars represent the S.D. of the transfection.

## Mass spectrometric analysis of melanopsin suggests proximal and distal C-terminal phosphorylation after short and prolonged light exposure, respectively

Previous studies on melanopsin's C-terminus phosphorylation state have used mutagenesis to propose the phosphorylation sites needed for melanopsin deactivation. Here, melanopsin-expressing HEK293 cells were exposed to either 1 min or 30 min of white light, quenched, and melanopsin was then affinity-purified for proteolytic digestion using trypsin or a trypsin/chymotrypsin combination (1 min light and dark samples were subjected to in-gel tryptic digest and 30 min light samples were subjected to trypsin/chymotrypsin digest in solution). We interrogated the phosphorylation state of melanopsin at the 1 min time point because in our HEK calcium imaging assays melanopsin has completed its deactivation at approximately 1 min. We thus predicted that we would observe a significant enrichment of phosphorylated melanopsin at this time. The 30 min time point was chosen to test for phosphorylation after prolonged light exposure. This light paradigm simulates the type of light exposure ipRGCs

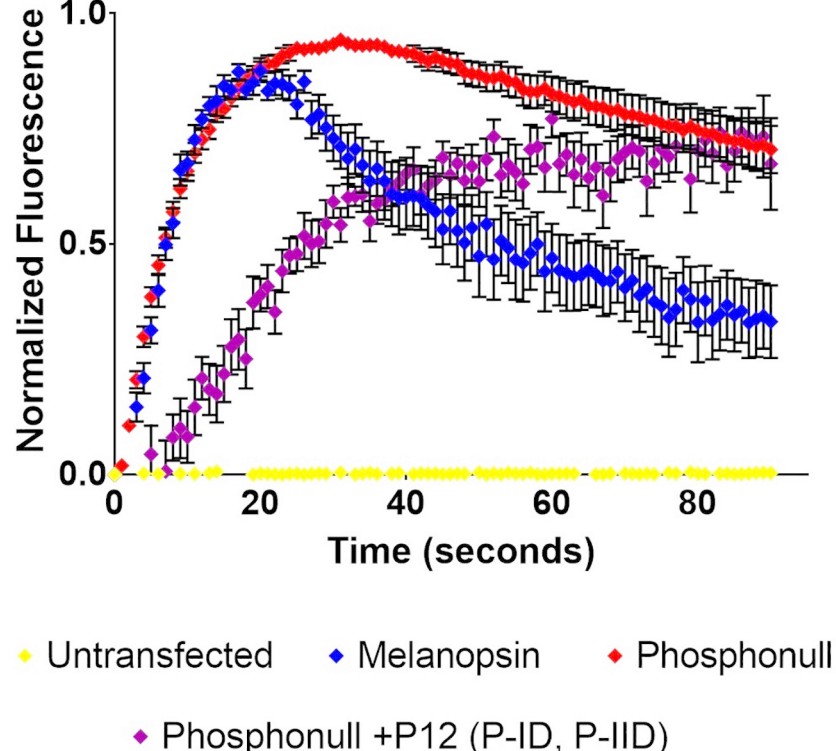

**Fig 6. Phospho-mimetic mutation of P-I and P-II in phosphonull melanopsin does not rescue deactivation defect.**
Phosphonull melanopsin was used to mutate P-I and P-II from alanine to aspartic acid to synthesize the phosphonull +P12 (P-ID, P-IID) mutant. Calcium imaging data show that this phospho-mimetic construct doesn't reach peak signaling activation in the time span of the assay. Error bars represent the S.E.M. of three independent transfections.

experience during the day. Proteolytic digestion resulted in ~30–40% sequence coverage for light-exposed melanopsin, the majority of which covers transmembrane helix 7 and the C-terminus (S3 Fig). Specifically, we observed C-terminal sequence coverage at P-I, P-II, and partially at P-III in 1 min light-exposed samples and we observed coverage at P-II, P-III, and P-IV in 30 min light-exposed samples. We observed low sequence coverage in the dark sample, with coverage only at P-I. We attribute this overall low protein expression yields after transient transfection of melanopsin's gene and limited access to proteolytic digestion sites due to the hydrophobicity of the transmembrane domains of the protein. Mass spectrometric analysis with high level (>80%) of sequence coverage has been obtained for few GPCRs, such as bovine rhodopsin and human cannabinoid 1 receptor [31]. Complete lists of b and y ions identified in mass spectrometric analysis are provided in Tables 1–6.

After 1 min of light exposure, three phosphopeptides were detected ($^{267}$ACEGCGEpSPLR$^{277}$, $^{376}$SHPpSLSYR$^{383}$/$^{376}$SHPSLpSYR$^{383}$, $^{376}$SHPpSLSYR$^{383}$/$^{376}$SHPSLpSYR$^{383}$/$^{376}$SHPSLSpYR$^{383}$), suggesting phosphorylation of residues S274, S379/S381, S379/S381/Y382 (Fig 7A, 7B and 7C), which are located in the proximal region of the C-terminus (P-I). The second and third phosphopeptides' phosphorylation sites could not be confidently assigned to a single amino acid(s), thus suggesting that both S379 and S381 could potentially be the residues phosphorylated in the second phosphopeptide and S379, S381, and Y382 could be phosphorylated in the third phosphopeptide.

Interestingly, mass spectrometric analysis after a 30-minute light exposure identified two phosphopeptides, distinct from the 1-minute light exposure

**Table 1. Complete list of b and y ions identified after mass spectrometric analysis of dark adapted melanopsin sample.**

Dark Sample—$^{376}$SHPSLpSYR$^{383}$

| Amino acid | # | b | y |
|---|---|---|---|
| S | 376 | 88.039 | - - - |
| H | 377 | **225.098** | 939.408 |
| P | 378 | **322.151** | 802.349 |
| S | 379 | **409.183** | 705.297 |
| L | 380 | 522.267 | **618.265** |
| S | 381 | **689.265** | 505.181 |
| Y | 382 | 852.329 | 338.182 |
| R | 383 | - - - | 175.119 |

**Table 2. Complete lists of b and y ions identified after mass spectrometric analyses of 1-minute light exposed melanopsin samples.**

1 Minute Light Sample—$^{267}$ACEGCGEpSPLR$^{277}$

| Amino acid | # | b | y |
|---|---|---|---|
| A | 267 | 72.044 | - - - |
| C | 268 | **232.075** | 1244.444 |
| E | 269 | **361.118** | 1084.413 |
| G | 270 | **418.139** | **955.370** |
| C | 271 | **578.170** | **898.349** |
| G | 272 | **635.191** | **738.318** |
| E | 273 | **764.234** | 681.297 |
| S | 274 | 931.232 | **552.254** |
| P | 275 | 1028.285 | **385.256** |
| L | 276 | **1141.369** | **288.203** |
| R | 277 | - - - | 175.119 |

($^{400}$ISGRKRQEpSLGSESEVGWTDTETTAAW$^{426}$ & $^{400}$ISGRKRQEpSLGpSESEVGWTDTE TTAAW$^{426}$), which suggests the phosphorylation of residues S408 and S411 (Fig 7D and 7E). These phosphorylated residues are in the P-III region of the C-terminus. Interestingly, P-II and P-III result in the fastest deactivation kinetics of melanopsin (Fig 4A), faster even than wild-type melanopsin. These results suggest a more defined role of the distal C-terminus in

**Table 3. Complete lists of b and y ions identified after mass spectrometric analyses of 1-minute light exposed melanopsin samples.**

1 Minute Light Sample—$^{376}$SHPpSLSYR$^{383}$/$^{376}$SHPSLpSYR$^{383}$

| Amino acid | # | b | y |
|---|---|---|---|
| S | 376 | 88.039 | - - - |
| H | 377 | **225.098** | 939.408 |
| P | 378 | **322.151** | 802.349 |
| S | 379 | **489.419** | 705.297 |
| L | 380 | **602.233** | 538.298 |
| S | 381 | **689.265** | 425.214 |
| Y | 382 | 852.329 | 338.182 |
| R | 383 | - - - | 175.119 |

**Table 4. Complete lists of b and y ions identified after mass spectrometric analyses of 1-minute light exposed melanopsin samples.**

1 Minute Light Sample—[376]SHPpSLSYR[383]/[376]SHPSLpSYR[383]/[376]SHPSLSpYR[383]

| Amino acid | # | b | y |
| --- | --- | --- | --- |
| S | 376 | 88.039 | - - - |
| H | 377 | **225.098** | 939.408 |
| P | 378 | **322.151** | **802.349** |
| S | 379 | **409.183** | **705.297** |
| L | 380 | 522.267 | 618.265 |
| S | 381 | **689.265** | **505.181** |
| Y | 382 | 852.329 | **338.182** |
| R | 383 | - - - | **175.119** |

melanopsin deactivation than shown previously based on C-terminally truncated constructs
[13, 14].

We also analyzed melanopsin samples obtained from dark-adapted transfected HEK293
cells, without any exposure to light. Mass spectrometric analysis of these samples identified

**Table 5. Complete lists of b and y ions identified after mass spectrometric analyses of 30-minute light exposed melanopsin samples.**

30 Minute Light Sample—[400]ISGRKRQEpSLGSESEVGWTDTETTAAW[426]

| Amino acid | # | b | y |
| --- | --- | --- | --- |
| I | 400 | - - - | - - - |
| S | 401 | 201.12 | 2948.3 |
| G | 402 | 258.14 | 2861.3 |
| R | 403 | 414.25 | 2804.3 |
| K | 404 | 542.34 | 2648.2 |
| R | 405 | 698.44 | 2520.1 |
| Q | 406 | 826.5 | 2364 |
| E | 407 | 955.54 | 2235.9 |
| S | 408 | 1122.5 | 2106.9 |
| L | 409 | 1235.6 | **1379.8** |
| G | 410 | 1292.6 | **1826.6** |
| S | 411 | 1379.7 | 1769.8 |
| E | 412 | 1508.7 | 1682.7 |
| S | 413 | 1595.8 | **1553.6** |
| E | 414 | **1724.6** | 1466.6 |
| V | 415 | 1823.9 | **1337.5** |
| G | 416 | **1880.7** | **1238** |
| W | 417 | 2067 | 1181.5 |
| T | 418 | 2168 | 995.43 |
| D | 419 | 2283 | **894.1** |
| T | 420 | 2384.1 | **779.3** |
| E | 421 | 2513.1 | 678.31 |
| T | 422 | 2614.2 | 549.27 |
| T | 423 | 2715.2 | 448.22 |
| A | 424 | 2786.3 | 347.17 |
| A | 425 | 2857.3 | 276.13 |
| W | 426 | - - - | 205.1 |

**Table 6. Complete lists of b and y ions identified after mass spectrometric analyses of 30-minute light exposed melanopsin samples.**

30 Minute Light Sample—$^{400}$ISGRKRQEpSLGpSESEVGWTDTETTAAW$^{426}$

| Amino acid | # | b | y |
|---|---|---|---|
| I | 400 | - - - | - - - |
| S | 401 | 201.12 | 3028.27 |
| G | 402 | 258.14 | 2941.24 |
| R | 403 | **414.1** | 2884.21 |
| K | 404 | 542.34 | 2728.11 |
| R | 405 | **698.9** | 2600.02 |
| Q | 406 | 826.50 | 2443.92 |
| E | 407 | 955.54 | 2315.86 |
| S | 408 | 1122.54 | 2186.82 |
| L | 409 | 1235.62 | 2019.82 |
| G | 410 | **1292.6** | 1906.73 |
| S | 411 | 1459.64 | **1849.1** |
| E | 412 | 1588.68 | 1682.71 |
| S | 413 | 1675.72 | **1553.7** |
| E | 414 | 1804.76 | 1466.64 |
| V | 415 | 1903.83 | **1337.4** |
| G | 416 | 1960.85 | **1238.5** |
| W | 417 | 2146.93 | **1181.2** |
| T | 418 | 2247.97 | 995.431 |
| D | 419 | 2363.00 | 894.38 |
| T | 420 | 2464.05 | 779.35 |
| E | 421 | 2593.09 | 678.30 |
| T | 422 | 2694.14 | **549.1** |
| T | 423 | 2795.19 | **448.1** |
| A | 424 | 2866.22 | 347.17 |
| A | 425 | 2937.26 | 276.13 |
| W | 426 | - - - | 205.09 |

one phosphopeptide ($^{376}$SHPSLpSYR$^{383}$) which suggests phosphorylation at residue S381 in the dark (Fig 7F). This finding is consistent with the previous results found in rat melanopsin using phosphoserine-specific antibodies which detected phosphorylation at that S381 in the dark [19].

## Structural modeling of melanopsin suggests proximal C-terminus phosphorylation sites (P-I & P-II) locate ideally for interaction with β-arrestin-1

Recent efforts to describe the structures of GPCR complexes and their formation has provided invaluable information about the molecular determinants necessary for binding and activation of signaling molecules. Most relevant to this study is the body of structural work investigating GPCR—C-terminus phosphorylation and its effect on arrestin recruitment, activation, and binding [32]. We present here a model of melanopsin in complex with β-arrestin-1 (Fig 8) to provide a structural model for the importance of melanopsin C-terminus' phosphorylation sites in arrestin recruitment, specifically through interaction with key regions on arrestin.

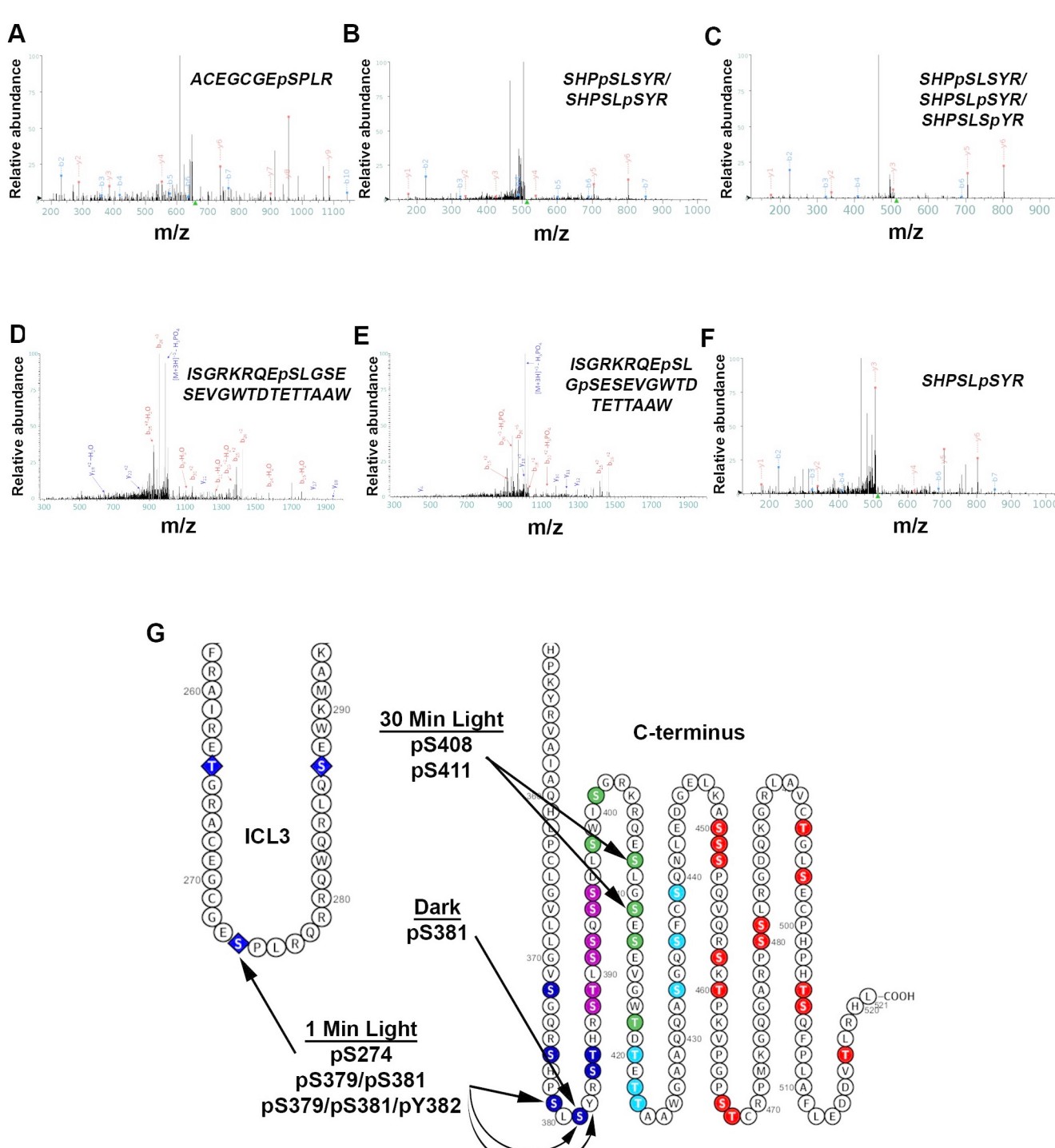

**Fig 7. Mass spectrometry of light-exposed melanopsin and dark-adapted melanopsin reveals C-terminus phosphorylation, and cytoplasmic loop phosphorylation.** Annotated fragmentation spectra of two phosphopeptides detected after LC/MS/MS of affinity purified melanopsin expressed in HEK293 cells, following a 1-min (A-C) and 30-min (D-E) white light exposure and after dark-adaptation with no light exposure (F). (A) Spectrum of phosphopeptide [267]ACEGCGEpSPLR[277], (B) [376]SHPpSLSYR[383]/[376]SHPSLpSYR[383], and (C) [376]SHPpSLSYR[383]/[376]SHPSLpSYR[383]/[376]SHPSLSpYR[383] suggest cytoplasmic loop 3 and proximal C-terminus phosphorylation following 1-minute of light exposure. Amino acids in the phosphopeptides in (B) and (C) were not confidently assigned to single amino acid(s), thus all the sites described in each phosphopeptide represent possible phosphorylations. (D) Spectrum of phosphopeptide [400]ISGRKRQEpSLGSESEVGWTDTETTAAW[426] and (E) spectrum of phosphopeptide [400]ISGRKRQEpSLGpSESEVGWTDTETTAAW[426] both suggest distal C-terminus phosphorylation following a prolonged, 30-min light exposure. (F) Spectrum of phosphopeptide [376]SHPSLSpYR[383] suggesting phosphorylation of S381 in the dark. (G) Summary of the findings on an abbreviated secondary structure of mouse melanopsin.

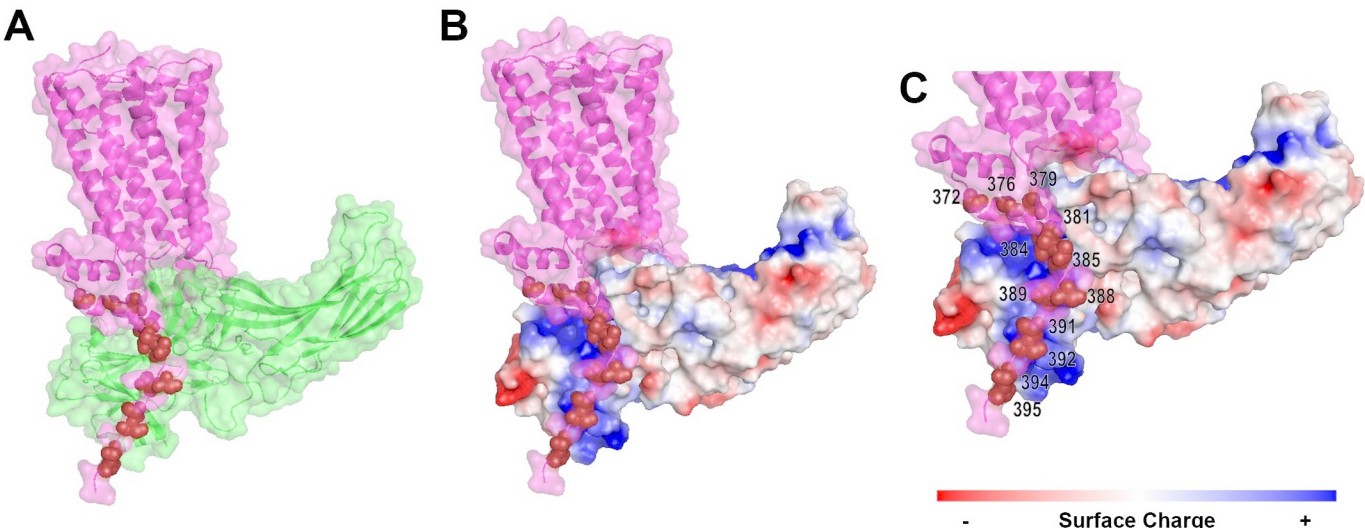

**Fig 8. Structural modelling of the melanopsin—β-arrestin-1 complex highlights the importance of proximal C-terminus phosphorylation sites.** (A) Melanopsin (pink) in complex with β-arrestin 1 (green). Proximal C-terminus phosphorylation sites are indicated in dark red. (B) Melanopsin (pink) in complex with β-arrestin 1, which has surface charge indicated by red, blue, or white. Proximal C-terminus phosphorylation sites indicated by light blue spheres. (C) Zoomed in view of (B), with phosphorylation sites numbered.

Our modeling suggests that melanopsin's proximal twelve C-terminus phosphorylation sites are optimally positioned for interaction with the major positively charged regions on arrestin. The positive regions on arrestin are critical for arrestin's detection of phosphorylated GPCRs and its subsequent activation and conformational change [33, 32]. Of the proximal twelve sites, our model suggests that the nine residues S381, S384, T385, S388, T389, S391, S392, S394, S395 are positioned optimally for arrestin interaction (Fig 8). Furthermore, we predict that the distal portion of the tail (not depicted in Fig 8), containing phosphorylatable residues after the proximal twelve sites, is less structured and ordered than the proximal C-terminus, thus limiting its ability to interact with the positively-charged phosphorylation-detection regions on arrestin. However, our functional and mass spectrometric data (Figs 4 and 7) suggest that distal C-terminal phosphorylation sites might also couple to arrestin, despite the possible lack of conformational or structural order of the C-terminus in this region. Taken together, our structural model suggests that phosphorylation of S388, T389, S391, S392, S394, S395 and the neighboring phosphorylatable amino acids around this cluster is critical to melanopsin's capability to attract, bind, and activate arrestins via interaction with arrestin's phosphorylation sensing domains.

## Discussion

The mouse melanopsin C-terminus has a uniquely large number of potential phosphorylation sites. Given our results reported here, we propose that melanopsin's C-terminus undergoes phosphorylation modification on a larger number of serine and threonine residues, centered primarily on a cluster of six sites, P-II (S388, T389, S391, S392, S394, S395), and additional phosphorylations of residues immediately surrounding it (P-I or P-III). While our data suggests that very distal C-terminus serine and threonine residues can deactivate melanopsin (P-IV), they do so at a reduced capability. We present data supporting the importance of amino acids in P-II that can be phosphorylated and select residues adjacent to it in either the proximal or distal direction, particularly S274, S379, S381, or Y382 after 1 min of light

exposure, S408 and S411 during prolonged light exposure, and S381 in the dark. These data therefore suggest melanopsin's C-terminus can exist in multiple light-dependent and dark-adapted phosphorylated states. This is not unexpected due to the predicted extended and unstructured nature of the C-terminus (past residue 396). The data presented here and past work [13, 14] suggest that deactivation is mediated by phosphorylation of proximal and distal C-terminus phosphorylation sites. Additionally, two phosphorylation sites were previously identified in rat melanopsin, S381 and S398, that are phosphorylated in the dark and light, respectively [19]. Previous calcium imaging analysis of melanopsin C-terminal serine and threonine point mutations in HEK293 cells [14], specifically, of P-II and three of the P-I sites (S381, S384, T385) reveals that mutation of these nine residues results in the most delayed deactivation, indicated as higher intensity light responses than wildtype melanopsin. However, this previous study [14] did not have a complete C-terminal phosphonull mutant for comparison, and calcium imaging assays were done with a single excitatory illumination as opposed to our study, which excites the cells with 487 nm light at a rate of 1 Hz. Thus, the work by Mure et al. [14] provides confirmation of the importance of proximal C-terminal phosphorylation sites, as found previously by Blasic et al. [13], while also providing in vivo evidence of their importance as well. We also present evidence of the importance of these proximal sites, all twelve of them (P-I & P-II), but also of the distal (after residue 396) C-terminal phosphorylation sites (P-III & P-IV), and also highlight the flexibility of melanopsin's deactivation, which can be achieved using proximal or distal sites. Taken together, we suggest that all of these data are not contradictory, instead we propose that these data all represent the various phosphorylation states of melanopsin's C-terminus. As our data suggest, various phosphorylation states of the C-terminus contribute to signaling deactivation.

The relatively long C-terminus of melanopsin raises some interesting structural questions. To mention a few, does the very distal region of the C-terminus, specifically, after residue S418, play any role in deactivating wild-type melanopsin? Our C-terminus mutant, Melanopsin +P12 (P-II, P-IV), suggests a limited role of the distal region in the deactivation of melanopsin, but does phosphorylation of these serine and threonine residues normally occur? We propose that distal putative phosphorylation sites can indeed be phosphorylated, even though no phosphorylation at these sites was detected through mass spectroscopy. Alternatively, there is the possibility that the serine and threonine residues in P-IV, and even in P-I, hold structural significance rather than just serving as sites of phosphorylation. These sites, whether phosphorylated or not, could help to maintain the C-terminus in a conformation that improves the accessibility of critical binding regions (i.e. intracellular loops) though the formation of hydrogen bonds or salt bridges at select serine and threonine residues on the C-terminus. Thus, this might be an interesting mechanism involved in modulating signaling activity. Specifically, through altering the degree of steric freedom at the melanopsin-signaling molecule complex interface. In the mouse, melanopsin also exists in two isoforms, OPN4S (short) and OPN4L (long), where the short isoform predominately exists in the developing retina, and long isoform expression increases as the mouse ages, past P14 [34, 35]. Opn4S possesses a shorter C-terminus than Opn4L, but they are identical to Opn4L up to residue Q454, thus, both isoforms possess P-I, P-II, P-III, and P-IV clusters of phosphorylation sites, which we show to be the primary regulators of melanopsin deactivation. Additionally, previous studies using electrophysiology or calcium imaging of transfected cells show that both isoforms possess light responses of identical amplitude and kinetics [13, 34]. Thus, it is unlikely that the melanopsin isoforms differ in their light-dependent phosphorylation and regulation of signal transduction, and if they do differ functionally, it would be through a separate mechanism not described here.

Taking all this into account, we propose a model of melanopsin C-terminus functional versatility where light-activation induces phosphorylation of the C-terminus at select residues on

P-I, P-II, and P-III, which produces signaling deactivation through arrestin activation and binding. Specifically, our findings suggest that P-II is the initial region of phosphorylation, likely by GRK, and our findings also suggest that additional phosphorylation occur on both P-I and P-III clusters. P-II phosphorylation also occurs very rapidly and is tightly regulated (i.e. rapid de-phosphorylation kinetics), as we did not capture P-II phosphorylation in 1 min of light exposure. Furthermore, P-III holds more importance for signal termination than P-I, as evidenced by the profound effects on light responses after its mutagenesis. P-IV can also be phosphorylated and have an impact on light response termination but holds the least functional significance out of all examined clusters. Thus, the time frame of reactions is proposed to occur as follows: dark-adapted melanopsin exists with P-I phosphorylation to prime the cytoplasmic conformation for G-protein activation. Then, after light illumination, P-II is rapidly phosphorylated, followed by P-III and P-I phosphorylation, and finally, phosphorylation of P-IV or distal sites. Simultaneous with these phosphorylation reactions, there is rapid coupling of melanopsin to signaling molecules (i.e. G-protein and arrestin) and simultaneously, there is rapid regulation of the phosphorylation state by de-phosphorylation of the most critical cluster, P-II. P-III phosphorylation persists after prolonged light exposure, suggesting that these sites also help regulate sustained light responses in addition to the rapid, initial deactivation response. Additionally, phosphorylated serine and threonine residues throughout the C-terminus regulate formation of signaling complexes and modulate melanopsin signaling at the level of the receptor.

While the P-II region is the most important and necessary cluster of potential phosphorylation sites, we didn't observe phosphopeptides with phosphorylations in this region during any of our time points in our mass spectrometric analysis. While unexpected, there is the possibility that phosphorylation of these sites is much more transient than at the sites we observed at 1 min or 30 min. Phosphorylation of P-II could be induced in a much more rapid timeframe as well, potentially immediately following light-activation of melanopsin, and this would also raise the possibility of dephosphorylation kinetics being rapid as well. Alternatively, the residues at P-II could serve as a binding site for either kinase or arrestin, instead of being direct sites of phosphorylation. While our data implicates P-I, -II, -III, and -IV in the regulation of melanopsin deactivation, it is also conceivable that some of these phosphorylations could be performing alternative and undescribed function. Given the length of the C-terminus, it might be possible that modification of very distal and unexplored residues could lead to interaction with binding partners that remain to be elucidated.

We also propose that our data suggests interesting mechanisms used in ipRGC phototransduction, given recent findings that suggest a diversity of signaling amongst different ipRGC subtypes. Specifically, electrophysiological data in mutant mice lacking Gαq signal transduction in ipRGCs suggest the presence of an additional, cyclic nucleotide-mediated signaling cascade primarily in M4 cells, and also in M2 cells [36]. Also, melanopsin Gαq-type phototransduction is modified in M4 cells to adjust the cell's excitability for contrast sensitivity across dim and bright light [37]. Additionally, the M1 subtype is not a homogenous cell population in its biophysical properties or transcription factors [38]—instead, cells within this subtype will vary in response to light intensity [39, 40]. Given our results, we propose the interesting possibility that melanopsin C-terminal phosphorylation could be re-purposed or adjusted to meet the needs of the given ipRGC, and potentially contribute to creating the physiological diversity observed amongst all ipRGCs. We are aware that our study was performed in a heterologous system (HEK293 cells), thus any direct comparisons or predictions to ipRGC function should be made conservatively. However, heterologous expression of melanopsin has been used in several studies using many cell lines [reviewed in 41], with several advantages and disadvantages associated with the various systems—the main difference being

expression yields for functional or spectral analyses, and protein purification. We are also aware that kinase and arrestin concentrations might differ in HEK293 cells and ipRGCs, however, HEK293 cells possess high mRNA levels for a variety of kinases including the GRK, protein kinase A, and protein kinase C families [42]. In addition, both HEK293 cells and ipRGCs possess elevated levels of β-arrestin 1 and 2 [42, 43], which bind phosphorylated melanopsin to quench signal transduction. Additionally, analysis of phosphonull melanopsin in HEK293 cells [12, 13] generated the hypothesis that ipRGC-driven electrophysiology and behaviors would display delayed deactivation, and subsequent *in vivo* studies using mice with C-terminal phosphorylation mutants support this hypothesis [14, 15], thus validating conclusions and predictions obtained from in vitro analysis of melanopsin function. Taking this into account, our study generates an interesting question: if regulation of the C-terminus is indeed contributing to the unique functional properties of ipRGC subtypes, what are the potential phosphorylation profiles of these cells? This might result in ipRGC subtypes possessing unique "phosphorylation fingerprints" that have direct and distinct functional consequences in each cell type.

Melanopsin's flexibility in signaling deactivation through the regulation of C-terminus phosphorylation may be a phenomenon shared amongst other GPCRs with similarly long C-termini bearing many phosphorylatable residues. Some notable GPCRs with similar or longer C-termini than melanopsin amongst the class A (rhodopsin) GPCRs are adrenoreceptors (α1A, α1B α1D, β1, β2), D1 and D5 dopamine receptors, serotonin receptors (2A, 2B, 2C, 6, and 7), EP4 prostanoid receptor, 2A adenosine receptor, and a large amount of orphan receptors [44]. Notably, these receptors, including melanopsin, possess partial or complete "phosphorylation codes," (three serine/threonines with one or two residues between them) at the distal ends of their C-termini. This motif is proposed to be a universal activator of arrestin—specifically through ideal coupling of the receptor's C-terminus phosphorylations to arrestin's positively-charged phosphorylation-sensing domains [32]. It is thus conceivable that receptors with long C-termini might deactivate similarly to melanopsin, where proximal C-terminus phosphorylations—occurring on a region of the C-terminus with a higher degree of structural order—are required to recruit arrestin. At the same time, distal C-terminus phosphorylations also can deactivate signaling, but their capability to do so is hindered due to the structural disorder of the distal C-terminus. We propose that these findings can be translatable to other similarly structured and unexplored GPCRs.

## Experimental procedures

### Mutagenesis of melanopsin gene

Mouse melanopsin coding sequence (NCBI accession: NM_013887.2) in the mammalian expression vector PMT3 was mutated using site-directed mutagenesis [45] creating a silent mutation that resulted in an internal Kpn1 restriction site. This Melanopsin-Kpn1 construct served as the template for all subsequent mutations, done through cassette or site-directed mutagenesis. Synthetic cassettes of the last 70 amino acids of mouse β2-adrenergic receptor (NCBI accession: NM_007420.3) and the last 44 amino acids of mouse angiotensin II type-1a receptor (accession: NM_177322.3) were used to construct melanopsin chimeric constructs. Primers and gene-fragments synthesized by IDT (Integrated DNA Technologies, Inc.). All melanopsin mutants were sequence verified (Genewiz).

### Cell culture and heterologous expression of melanopsin

HEK293 cells (ATCC) were plated and adhered to dishes containing DMEM—high glucose, pyruvate (ThermoFisher Scientific) supplemented with 10% (v/v) fetal bovine serum

(ThermoFisher Scientific) and 1% (v/v) antibiotic-antimycotic (ThermoFisher Scientific) in a humid 5% $CO_2$ incubator set at 37 ˚C. Cells were passaged by disassociation using trypsin-EDTA (0.25%) (ThermoFisher Scientific), and seeded onto new plates with fresh supplemented DMEM.

Cells were transiently transfected with the pMT3 vector containing mouse melanopsin coding sequence as per Turbofect Transfection Reagent protocol (ThermoFisher Scientific) for preparation for calcium imaging assays. Briefly, 240,000 cells were seeded into each well of a 6-well culture plate (Corning). Vector DNA was diluted in DMEM and then Turbofect Transfection Reagent was added. After incubation, the DNA mixture was added to the cells and incubated overnight. The cells were then trypsin released and seeded onto a 96-well culture dish (Corning), 100,000 cells per well. The cells were then incubated in a $CO_2$ incubator overnight for dark-adaptation.

Transient expression of melanopsin for protein purification was carried out in a similar manner as for calcium imaging preparation. Briefly, $1x10^6$ cells were seeded onto 10 cm culture plates (Corning), and pMT3 vector containing melansopsin-1D4 (C-terminus epitope tag consisting of the last nine amino acids of bovine rhodopsin) was prepared in DMEM and Turbofect Transfection Reagent and then added to the cells. Transfected 10 cm plates were then stored in a light-safe $CO_2$ incubator for dark adaptation. 48 h later, in a darkroom under dim red-light illumination, transfected cells were harvested, and washed twice with PBS with 1X Halt Protease and Phosphatase Inhibitor Cocktail (ThermoFisher Scientific). After 1 min white light-exposure, 30 min white light exposure, or no light exposure, cell pellets were flash frozen in dry ice with ethanol and cell pellets were stored at -80 ˚C.

## Calcium imaging of transfected HEK293 cells

Due to melanopsin coupling to a Gαq signaling cascade, we utilized a fluorescent calcium dye to track fluctuations in intracellular calcium as a result of melanopsin phototransduction activation and deactivation. Calcium imaging assays were done in a dark room under dim red-light illumination using the Fluo-4 Direct adherent cell protocol (ThermoFisher Scientific). Briefly, 96-well plates containing transiently transfected and dark-adapted HEK293 cells were dispensed of their growth media and replaced with a 1:1 mixture of supplemented DMEM and Fluo-4 Assay Reagent (containing 5 mM probenecid). Cells were also incubated with 20 µM 9-*cis*-retinal (Sigma) to reconstitute melanopsin. After a 1 h incubation, the media was dispensed and replaced with HBSS (Corning) containing 20 mM HEPES (Sigma). Melanopsin signaling kinetics were then measured using a TECAN Infinite M200 (TECAN Trading AG) by exciting the sample at 487 nm and recording the emission fluorescence at 516 nm at a rate of 1 Hz. Experiments were carried out in replicates of six per melanopsin construct. The excitation light simultaneously activates melanopsin and the Fluo-4 AM dye, thus melanopsin is continuously excited for the duration of the experiment. Constant excitation with 487 nm light should produce melanopsin's photoproduct (metamelanopsin), but should not drive the photoproduct back to an inactive state, since a higher wavelength light pulse is needed to drive this process [7]. Data were normalized to facilitate rate calculation (see next section), in case of differences in absolute fluorescence attributable to cell passage number. In each transfection, data for each construct was normalized to its own maximal fluorescence, then data from multiple transfections were pooled together for rate calculation and statistical analyses. We did not observe striking differences in absolute fluorescence amongst mutants in individual transfections, and differences in melanopsin expression affects absolute fluorescence, but do not alter normalized deactivation kinetics (S5 Fig).

## Calculation of melanopsin deactivation rate

The deactivation phases of calcium imaging assay data (corresponding to the part of the normalized data after the peak fluorescence level) for all melanopsin constructs was fitted to an exponential decay function using GraphPad Prism software (GraphPad Software, Inc.). The following function was used:

$$y = y(0) * e^{(k*x)}$$

Where $y(0)$ is the value at $t = 0$, and k is the rate constant. Data across multiple transfections (six replicates per transfection) were pooled together and averaged, and the deactivation rate of the averaged data was calculated. Standard error of the mean of the deactivation rates of all melanopsin constructs and statistical analysis were calculated using GraphPad software. Statistical significance (*, **, ***, **** denotes $p < 0.5, 0.01, 0.001, 0.001$, respectively) was determined by performing unpaired t-tests of mutant melanopsin constructs with respect to wild-type melanopsin, or with another construct of interest (indicated in each figure legend). Corrections for multiple comparisons was done using the Holm-Sidak method on GraphPad.

## Affinity purification of melanopsin

We used a modified version of a previously described protocol [46]. Briefly, we immobilized monoclonal 1D4 antibody (generously provided by D. Oprian, Brandeis University) onto Sepharose-4B (GE Healthcare Life Sciences). All steps were done in a darkroom under dim red-light illumination. Transfected HEK293 cells were solubilized in 0.1–1% (w/v) n-dodecyl-β-D-maltoside (ThermoFisher Scientific), 1 mM phenylmethylsulfonyl fluoride (Sigma-Aldrich), 50 mM HEPES (Sigma-Aldrich), 3 mM $MgCl_2$, 140 mM NaCl, 1X Halt Protease and Phosphatase Inhibitor Cocktail (ThermoFisher Scientific) at 4 °C. Solubilized protein was then incubated with Sepharose-4B conjugated with 1D4 antibody at 4 °C. The mixture was then centrifuged, the supernatant discarded, and the Sepharose-4B-1D4-antibody resin was resuspended in solubilization buffer and then packed into disposable spin columns to facilitate a series of 20 washes with solubilization buffer. Melanopsin protein bound to the antibody resin was eluted by incubating with solubilization buffer with 50 μM 1D4 peptide (amino acid sequence TETSQVAPA). The eluate was spun using Amicon Ultra 10K Centrifugal Filters (Millipore Sigma) to concentrate the purified melanopsin protein and filter out the eluting peptide.

## Mass spectroscopy of melanopsin purified from HEK293 cells

The following protocol was used for 30-min light exposed samples. Purified melanopsin protein was treated to 100 ng of trypsin and chymotrypsin in-solution, individually, as per the manufacturer's protocol (Promega). Digestion was carried out overnight at 37 °C. Peptides were then desalted using Pepclean C-18 spin columns (ThermoFisher Scientific) as per the manufacturer's protocol and then resuspended in 1% (v/v) acetic acid to a final volume of 30 μL. Small aliquots (5 μL) of purified, proteolytically-treated melanopsin was injected into a Fusion Lumos tribrid mass spectrometer system (Thermo Scientific). The HPLC column was a Dionex 15 cm x 75 μm Acclaim Pepmap C18, 2μm, 100 Å reversed- phase capillary chromatography column. Peptides were eluted with a gradient from 98% of buffer A containing 0.1% (v/v) formic acid in H2O and 2% of buffer B containing 0.1% (v/v) formic acid in acetonitrile to 2% of buffer A and 98% of buffer B at a flow rate of 0.3 μL/min. The microelectrospray ion source was operated at 2.5 kV. Data were analyzed by using all CID spectra collected in the experiment to search whole mouse UniProtKB databases with the search program Sequest HT

bundled into Proteome Discoverer 2.2 (ThermoFinnigan). A false discover rate of 1% at protein and peptide levels was used.

For 1-min and dark melanopsin samples, concentrated eluates from affinity purification were size separated through SDS-PAGE electrophoresis followed by Coomassie staining. After excising the desired bands, excised gel bands were cut into approximately 1 mm$^3$ pieces. The samples were reduced with 1 mM DTT for 30 min at 60 ˚C and then alkylated with 5 mM iodoacetamide for 15 min in the dark at room temperature. Gel pieces were then subjected to a modified in-gel trypsin digestion procedure [47]. Gel pieces were washed and dehydrated with acetonitrile for 10 min. followed by removal of acetonitrile. Pieces were then completely dried in a speed-vac. Rehydration of the gel pieces was with 50 mM ammonium bicarbonate solution containing 12.5 ng/μl modified sequencing-grade trypsin (Promega) at 4 ˚C. Samples were then placed in a 37 ˚C room overnight. Peptides were later extracted by removing the ammonium bicarbonate solution, followed by one wash with a solution containing 50% acetonitrile and 1% formic acid. The extracts were then dried in a speed-vac (~1 hr). The samples were then stored at 4 ˚C until analysis. On the day of analysis, the samples were reconstituted in 5–10 μl of HPLC solvent A (2.5% acetonitrile, 0.1% formic acid). A nano-scale reverse-phase HPLC capillary column was created by packing 2.6 μm C18 spherical silica beads into a fused silica capillary (100 μm inner diameter x ~30 cm length) with a flame-drawn tip. After equilibrating the column each sample was loaded via a Famos auto sampler (LC Packings) onto the column. A gradient was formed, and peptides were eluted with increasing concentrations of solvent B (97.5% acetonitrile, 0.1% formic acid). Each peptide was eluted, subjected to electrospray ionization, and then entered an LTQ Orbitrap Velos Pro ion-trap mass spectrometer (ThermoFisher Scientific). Eluting peptides were detected, isolated, and fragmented to produce a tandem mass spectrum of specific fragment ions for each peptide. Peptide sequences (and hence protein identity) were determined by matching protein or translated nucleotide databases with the acquired fragmentation pattern by the software program Sequest (ThermoFinnigan). The modification of 79.9663 mass units to serine, threonine, and tyrosine was included in the database searches to determine phosphopeptides. Phosphorylation assignments were determined by the Ascore algorithm [48]. All databases include a reversed version of all sequences and the data were filtered to between a one and two percent peptide false discovery rate.

## Structural modeling of melanopsin

Homology modeling of melanopsin was done using LOMETS, an online service for protein structure prediction [49]. The mouse melanopsin structural model used in this study was constructed using squid rhodopsin (PDB ID 2ZIY) [50] as the template. In order to model the proper orientation of arrestin when in complex with melanopsin, melanopsin was aligned with the structure of human rhodopsin in complex with mouse visual arrestin (PDB ID 4ZWJ) [33] using the cealign tool. Then, the structure of rat β-arrestin 1 in complex with V2 vasopressin receptor phosphopeptide (PDB ID 4JQI) [51] was aligned, using cealign, onto visual arrestin to orient β-arrestin 1 properly onto the intracellular face of melanopsin. The resulting structural model is a prediction of β-arrestin and melanopsin in complex, and the orientation of melanopsin's proximal C-terminus onto the phosphorylation-sensing domains on arrestin.

## Accession codes

*Mus musculus* OPN4: NCBI NM_013887.2

*Mus musculus* ADRB2: NCBI NM_007420.3

*Mus musculus* AGTR1A: NCBI NM_177322.3

## Supporting information

**S1 Fig. P-I is insufficient for signaling deactivation.** Calcium imaging of phopshonull + P6 (P-I sites mutated from alanine to serine and threonine residues). Error bars represent S.E.M. of three transfections.
(TIF)

**S2 Fig. Putative phosphorylation sites in P-I that are closer to P-II hold higher functional significance than P-I sites located further upstream on the C-terminus.** Calcium imaging of melanopsin phosphonull mutants with P-II mutated from alanine to serine and threonine residues and additional four sites on P-I mutated to serine and threonine residues. Mutant with P-II, and the four residues immediately before it mutated to serine and threonine residues (A379S, A381S, A384S, A385T) deactivates at the fastest rate among these mutants. Conversely, the mutant with P-II and the four additional residues furthest upstream on the C-terminus mutated to serine and threonine residues (A372S, A376S, A379S, A381S) displays the slowest deactivation rate, similar to phoshonull melanopsin. Error bars denote S.E.M. of three transfections.
(TIF)

**S3 Fig. Sequence coverage of proteolytically-digested melanopsin after mass spectrometry analysis.** Melanopsin sequence coverage based on detected peptides following mass spectrometry denoted on its amino acid sequence. Sequence coverage following 1 min white light exposure (A), 30 min white light exposure (B), and after dark adaptation with no exposure to light (C).
(TIF)

**S4 Fig. Amino acid sequence alignment of various mammalian melanopsins.** Amino acid alignment of several mammalian melanopsins, to highlight the conservation of the C-terminal phosphorylation sites, particularly the proximal sites. Colors on the amino acids correspond to the residue's property (Yellow: Non-polar, Green: Polar, Red: Negatively-charged, Blue: Positively-charged). C-terminus region highlighted on consensus sequence. Alignment done using Geneious software.
(TIF)

**S5 Fig. Titration of melanopsin transfections indicate that differences in melanopsin expression does not alter normalized signaling kinetics, only overall fluorescence.** HEK293 cells were transfected with various amounts (in μg) of PMT3 plasmid containing mouse melanopsin's gene. (A) Calcium imaging of varying concentrations of melanopsin, indicating absolute fluorescence intensities (in arbitrary fluorescence units, a.u.). Overall intensities increase as more melanopsin is transfected. (B) Normalized depiction of data in A. Note that signaling kinetics, specifically, the rate of activation and inactivation, are not affected by amounts of melanopsin transfected.
(TIF)

## Acknowledgments

We would like to thank Prof. Mark R. Marten and Cynthia L. Chelius for guidance, training, and help with mass spectrometry protein preparation. We would also like to thank Ross Tomaino and the Taplin Mass Spectrometry Facility, Cell Biology Department, Harvard Medical School, for their assistance with mass spectrometric analysis of melanopsin samples. We also thank Prof. Daniel D. Oprian for generously providing 1D4-antibody. The authors also acknowledge Haya AlGrain for advice during the writing of the manuscript. Gatan Inc. is the

current location for author SG, and data collection was done while affiliated with the University of California, Irvine. The specific roles of this author are articulated in the "Author Contributions" section.

## Author Contributions

**Conceptualization:** Juan C. Valdez-Lopez.

**Data curation:** Juan C. Valdez-Lopez, Sahil Gulati.

**Funding acquisition:** Phyllis R. Robinson.

**Investigation:** Elelbin A. Ortiz.

**Supervision:** Phyllis R. Robinson.

**Visualization:** Juan C. Valdez-Lopez.

**Writing – original draft:** Juan C. Valdez-Lopez.

**Writing – review & editing:** Krzysztof Palczewski, Phyllis R. Robinson.

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
