## [Decision Letter · Decision Letter 0]

25 Feb 2020

PONE-D-20-00450

Melanopsin Carboxy-terminus Phosphorylation Plasticity and Bulk Negative Charge, not Strict Site Specificity, Achieves Phototransduction Deactivation

PLOS ONE

Dear Dr Robinson,

Two experts in the field reviewed this manuscript. In general, they both supported the paper, and offered a number of suggestions and comments. After careful consideration, we feel that it has merit, but does not quite fully meet PLOS ONE’s publication criteria as it currently stands. Based on the two reviews, we recommend that this paper undergoes a semi-major revision, where the authors address comments from both reviewers, especially the major comments outlined by Reviewer 1. Thus, we invite you to submit a revised version of the manuscript that addresses the points raised during the review process.

We would appreciate receiving your revised manuscript by Apr 10 2020 11:59PM. To enhance the reproducibility of your results, we recommend that if applicable you deposit your laboratory protocols in protocols.io, where a protocol can be assigned its own identifier (DOI) such that it can be cited independently in the future. For instructions see: http://journals.plos.org/plosone/s/submission-guidelines#loc-laboratory-protocols

We look forward to receiving your revised manuscript.

Kind regards,

Wayne Iwan Lee Davies, PhD

Academic Editor

PLOS ONE

Journal Requirements:

2. Thank you for stating the following in the Competing Interests section: No competing interest"

We note that one or more of the authors are employed by a commercial company: Gatan Inc.

3. Please upload a new copy of Supplemental Figure 4 as the detail is not clear. Please follow the link for more information: http://blogs.PLOS.org/everyone/2011/05/10/how-to-check-your-manuscript-image-quality-in-editorial-manager/

Reviewers' comments:

Reviewer's Responses to Questions

**Comments to the Author**

1. Is the manuscript technically sound, and do the data support the conclusions?

Reviewer #1: Partly

Reviewer #2: Yes

2. Has the statistical analysis been performed appropriately and rigorously? 

Reviewer #1: Yes

Reviewer #2: Yes

3. Have the authors made all data underlying the findings in their manuscript fully available?

Reviewer #1: Yes

Reviewer #2: Yes

4. Is the manuscript presented in an intelligible fashion and written in standard English?

Reviewer #1: Yes

Reviewer #2: Yes

5. Review Comments to the Author

Reviewer #1: This manuscript investigates the possible correlation between the manifold potential C-terminal phosphorylation sites in mouse melanopsin and the desensitization of its light-activated state. To this purpose the authors embark upon an impressive collection of experimental approaches, including site-directed mutagenesis, second messenger monitoring, MS-based peptide mapping and in-silico modeling. Involvement of phosphorylation of C-terminal sites in desensitization of GPCRs is well documented, and there is variable evidence this is also the case for melanopsins. This manuscript focuses on identification of (groups of) responsible sites. The authors generate novel insight, but the overall outcome is a bit disappointing. Very specific conclusions cannot be drawn, and the discussion section contains redundant elements, and lacks appropriate inclusion of literature data.

Major comments:

1. As far as I could find, two gene lineages of melanopsins are known, OPN4x and OPN4m. Mammals only contain the m-type, but also may generate splice variants. For mouse melanopsin a short and a long splice variant were reported (Pires (2009) J Neurosci) and this manuscripts studies the long one. This should be clearly mentioned in the text and the method section. In the discussion it could be speculated whether the results obtained for the long variant would also hold for the short version, since that might also contribute to the diversity in the ipRGC population.

2. There is considerable overlap with the approach in ref. 14 (Mure (2016) Neuron). That paper also investigates the contribution of phosphorylation of potential sites to desensitization of mouse melanopsin-L, and includes in-vivo analysis of transgenic mice. It is essential to the quality of the discussion section and in fact the entire paper that results in ref. 14 are compared with those reported in this manuscript, to be concluded by an integrative discussion.

3. A fair number of host cells have been used for the heterologous expression of melanopsins (see e.g. compilation in Shirzad (2016) Prog Retin Eye Res). HEK cells generate low expression levels but seem appropriate for analyzing signaling pathways. However, considering the large population of potential phosphorylation sites, and the alleged involvement of a fair number of kinases, it is questionable whether HEK cells form a proper representative for ipRGCs, since the kinase population and kinase and arrestin levels will vary between cell types. Hence, both the activation level, the kinetics of de-activation and the effect of mutations will differ in HEK cells compared to ipRGCs and probably even between ipRGC classes. Direct translational interpretations therefore are out of proportion. This element is briefly touched in the discussion, but should be further embraced, possibly also in the light of the in-vivo data in ref. 14.

4. Fluorescent monitoring of the Ca2+ level in cells is a good qualitative indicator for activation and desensitization of melanopsins. However, would it also allow the subtle quantitative comparison between WT and mutants, as executed in this manuscript ? For instance, the 487 nm excitation for the Ca2+ sensor will also excite melanopsin and metamelanopsin, thereby altering the photo-equilibrium between these states during the desensitization process, probably affecting the desensitization kinetics. The latter will then also depend on the expression level of functional melanopsin. Has it e.g. been assessed whether activation and deactivation kinetics are similar for different expression levels of WT melanopsin ? If the absorbance bands of melanopsin and metamelanopsin, and the functional expression levels somewhat differ between WT and mutants, this will affect the activation and deactivation kinetics, and impair quantitative comparison between all these species. These elements also need to be properly discussed.

5. Actually, the mass spectroscopic analysis is the most complex, but also most elegant experimental approach, since it exploits the WT protein and does not depend on mutational perturbation. Unfortunately, the shortest time interval undertaken is 1 min, in which the deactivation process has largely been completed. Has a shorter time interval not been considered ?

Minor comments:

6. Top sentence in the legend of Fig. 3 is incomplete ("At least eight proximal C-terminus phosphorylation sites in the are needed to deactivate melanopsin").

7. Re lines 245-247: Possibly the phosphomimetic mutant more rapidly, but reversibly, recruits arrestins, thereby slowing down activation ?

8. Tables 5 and 6 are necessary, but too distractive in the main text. I suggest to transfer these to the supporting information.

9. line 455: though -> through ?

10. line 525: "Small aliquots (5 μL) of purified melanopsin was injected ...". I presume this was after proteolytic treatment ?

11. line 534: "concentrated eluates were size separated....". These are eluates form the affinity purification column ?

12. Figure 2A: Why is the fluorescent intensity in this and subsequent figures normalized ? I presume, that many mutants will reach higher maximal intensity. This also is useful information and the corresponding deactivation rates are presented anyway in the corresponding column figure. At least, if I understand this properly from the methods section, the deactivation rates are calculated from the absolute data ?

13. Figure 3A: This figure a.o. shows the phosphonull + P12 mutant, containing P-I + P-II phosphorylatable sites. However in Fig. 4A phosphonull + P12 apparently stands for P-II + P-III sites, in Fig. 4B phosphonull + P12 apparently stands for P-II + P-IV sites, and in Fig. 4C phosphonull + P12 apparently stands for P-II + P-III sites, as well as for P-II + P-IV sites. Please explain or correct.

14. Figure 3B: This figure lacks the activation signal. Were the first 20 seconds or so cut off ? Maybe include these on a smaller time-scale. Also, many curves are hardly distinguishable. Either use off-sets or absolute intensities.

15. Figure 4C: The deactivation rate for the phosphonull mutant + P-II + P-III sites is very much larger compared to WT. However, this is absolutely not obvious from Fig. 4A. The main differences in Fig. 4A seem to be a somewhat slower activation rate (but only if the absolute maximal intensities are similar) and maybe a somewhat lower final level (although probably not significant different). Please explain.

16. Figure 5: Diamond shaped residues in the "foreign" C-terminals are not easily detected. Better diamond-shape the other residues and encircle the potential phosphorylation sites.

17. Figure 6: How was the curve of the phosphonull + P12 mutant (P-ID + P-IID) normalized ? Here it certainly is much more informative to show absolute intensities.

18. Figure 7: The fragmentation spectra are not very clear to me. Maybe include one-page size spectra in the supporting information. What is the meaning of the bold lines in the spectra ? Do these arise from the corresponding peptides ?

19. Figure S4: I could not find a reference to this figure in the text. If you want to use it still, I suggest to only display the relevant C-terminal sequences.

Reviewer #2: This work from Valdez-Lopez and colleagues systematically interrogates the location, mechanism, and role of phosphorylation of the C-terminus of melanopsin in deactivation. As expected from this group, the experiments are well-designed and test both necessity and sufficiency of phosphorylation at given locations. The authors report some very interesting findings and will be of wide interest to the field. I have only a couple, relatively minor, critiques.

1. The last line in the abstract states, “This degree of functional versatility could help explain the diverse ipRGC light responses as well as non-image and image forming behaviors, even though all six subtypes of ipRGCs express the same melanopsin gene Opn4.”

This seems like an overly sweeping claim. For example, much of the diversity between subtypes is now thought to result from different downstream targets of the melanopsin phototransduction cascades (Jiang et al., 2018 and Sonoda et al., 2018). It also glosses over the fact that there are two splice isoforms of Opn4 that are differentially expressed by ipRGC subtypes. Finally, the phrasing of “as well as non-image and image forming behaviors” makes the meaning of this phrase unclear, are the authors stating that the functional versatility of the melanopsin protein could explain the diversity of visual behaviors driven by ipRGCs? Again, this is too broad a claim because the ipRGC role in behavior is also determined by a host of other factors, not least the differential projection patterns of ipRGC subtypes, as well as their intrinsic physiological properties.

In short, this sentence should be removed or modified.

2. Some of the text in the figures is almost microscopic, including graph labels in, for example, Figure 4 (but also elsewhere in general) and including labeling of protein structure in, for example, Figure 5 (expecially in A-D). If the text in Figure 5 on the protein structures is not important, it should be removed, otherwise, please enlarge it.

6. PLOS authors have the option to publish the peer review history of their article (what does this mean?). If published, this will include your full peer review and any attached files.

Reviewer #1: No

Reviewer #2: No

---

## [Author Response · Author response to Decision Letter 0]

4 Mar 2020

3/4/20

To the Editors and Reviewers,

PLOS ONE

1160 Battery Street

Koshland Building East, Suite 225

San Francisco, CA 94111

We are submitting a revised version of the manuscript entitled “Melanopsin Carboxy-terminus Phosphorylation Plasticity and Bulk Negative Charge, not Strict Site Specificity, Achieves Phototransduction Deactivation” with authors: Juan C. Valdez-Lopez, Sahil Gulati, Elelbin A. Ortiz, Krzysztof Palczewski, and Phyllis R. Robinson. The authors would like to express their gratitude and appreciation to the editors and reviewers for their time, consideration, and constructive critique of our manuscript. We have made changes based on the comments received. In this cover letter we address all the concerns raised in the initial review. Below you’ll find our response to all major and minor points raised in the review They are addressed in numerical order below as they appeared in the review. Note comments by the reviewer are in italics.

Reviewer #1:

Major comments:

1) The reviewer raised concern about the two mouse splice variants of melanopsin, OPN4S and OPN4L. We addressed this point in the revised paper, as suggested by the reviewer. Briefly, work from our lab and others indicate that both splice variants produce similar light response intensity and kinetics, as measured by electrophysiology and calcium imaging. Both splice variants also possess the important phosphorylation sites (P-I, P-II, P-II, & P-IV) investigated in this study. Thus, both splice variants are essentially identical in their signal transduction capabilities, and if they differ, its through functions not detailed in this work.

2) The reviewer requested that we discuss our results in reference to Mure et al 2016. Discussion of this work was added in the discussion as suggested by the reviewer.

3) The reviewer was concerned about the use of HEK 293 cells as an expression system in this study. The concern about the use of HEK293 cells as an expression system and a note about how results in HEK cells translates to ipRGCs is now addressed in the discussion section of the revised manuscript as suggested by reviewer. Our approach of using heterologous expression systems to perform structure function studies of melanopsin have proven invaluable. Our recent in vivo study (Somsaundaram et al 2017) validate this approach, demonstrating reactions found in HEK cells translate to an in vivo mouse model

4) The reviewer expressed concern about the calcium assay used in this study. A discussion about of our assay has been added to the Methods Section to address the reviewers concern. It should be noted that the rate of deactivation is not a function of the amount of melanopsin expression over the range used in our studies.

5) The reviewer was wondering about the kinetics of the mass spec experiments included in this paper. This was addressed in the revised manuscript in the result section. 

Minor comments:

6) Figure legend 3 is incomplete. Sentence edited as suggested.

7) Lines 245-247. Commentary about phospho-mimetic mutant was edited as suggested.

8) Reviewer suggests Tables 5&6 are distracting in the main text. We believe that the ion tables for all time points should remain displayed together, and the 30 min tables should remain with the other time points. The authors are willing to move all tables to the supporting information, if advised to do so. 

9) Line 455 typo pointed out. Typo edited.

10) Reviewer asked for clarification of methods. The reviewer is correct, the 5 uL aliquots are proteolytically treated samples. Manuscript was changed to improve clarity.

11) Reviewer asked for clarification of methods The reviewer is correct the eluates were from the affinity purification column, the manuscript was edited for clarity.

12) The reviewer asked for clarification of how we handle data from the calcium imaging assay. This is now addressed at length in revisions made to the Methods section under “Calcium imaging”.

13) The reviewer asked for clarification of figure 3A.

Phosphonull + P12 just refers to a mutated version of the phosphonull melanopsin mutant with twelve sites mutated to serine and threonine residues, as in wildtype melanopsin. The numbers immediately after the +P12 indicate which Ser/Thr clusters were mutated. Thus, phosphonull +P12 is used for several mutants, depending which sites were mutated. The figure was edited to avoid any confusion (the +P12 was 

removed, and each mutant now only indicates which clusters were mutated, e.g. phosphonull +P-II & P-III)

14) The reviewer asked for clarification of figure 3B. The figure was edited to indicate that activation was omitted (X-axis starts later than t=0). This was done to facilitate viewing of deactivation of the various mutants. The diamond icons were enlarged to facilitate discrimination of mutants on the graph. 

15) The reviewer asked for clarification of the data in fig 4C and 4 A. Commentary about this point was made in the results section of this data.

16) The reviewer asked for a change in symbols in fig 5. The figure was edited to facilitate discrimination of the “foreign” C-terminal Ser & Thr residues. 

17) The reviewer asked for clarification regarding the normalization method for fig. 6 More information about normalization was added in Methods section. 

18) The reviewer wanted a clarification of fig. 7 Figure 7 was edited to facilitate clarity. Authors are also not sure what the reviewer meant by “bold lines in the spectra.” All peaks in the spectra have the same line thickness, and thus any “bold” appearances are only indicative of a higher concentration of spectral peaks at a given m/z region.

19) The reviewer questioned the inclusion of Fig. S4. Figure S4 is referenced in the manuscript, see the end of the manuscript. The figure legend was expanded to comment on justification of aligning many mammalian melanopsins.

Reviewer #2:

Major comments:

1) Reviewer 2 took issue with the last sentence of the abstract. This sentence was edited to improve clarity of the point we want to make. 

2) Reviewer 2 took issue with the text in some of the figures. All figures with small text were edited to enlarge them, and facilitate the viewing of text, as suggested by the reviewer. 

We appreciate your consideration of our manuscript for publication in PLOS ONE. We hope that the edits we made, based on the reviewers’ critiques, meet your standards for publication in your journal. We thank you for your time, and we hope to publish soon in PLOS ONE.

Sincerely,

Phyllis R. Robinson

Juan Valdez-Lopez

---

## [Editor Report · Decision Letter 1]

9 Mar 2020

Melanopsin Carboxy-terminus Phosphorylation Plasticity and Bulk Negative Charge, not Strict Site Specificity, Achieves Phototransduction Deactivation

PONE-D-20-00450R1

Dear Dr. Robinson,

We are pleased to inform you that your manuscript has been judged scientifically suitable for publication and will be formally accepted for publication once it complies with all outstanding technical requirements.

With kind regards,

Wayne Iwan Lee Davies, PhD

Academic Editor

PLOS ONE

Additional Editor Comments:

Thank you for submitting a revised version. I am happy that all the reviewers' concerns and comments have been addressed in this excellent paper.

---

## [Editor Report · Acceptance letter]

12 Mar 2020

PONE-D-20-00450R1 

Melanopsin Carboxy-terminus Phosphorylation Plasticity and Bulk Negative Charge, not Strict Site Specificity, Achieves Phototransduction Deactivation 

Dear Dr. Robinson:

I am pleased to inform you that your manuscript has been deemed suitable for publication in PLOS ONE. Congratulations! Your manuscript is now with our production department. 

With kind regards,

on behalf of

Dr Wayne Iwan Lee Davies 

Academic Editor

PLOS ONE